# The Effect of Periodontitis on Body Size Phenotypes in Adults without Diagnosed Chronic Diseases: The Korean National Health and Nutrition Examination Survey 2013–2015

**DOI:** 10.3390/ijerph21091180

**Published:** 2024-09-04

**Authors:** Young Sang Lyu, Youngmin Yoon, Jin Hwa Kim, Sang Yong Kim

**Affiliations:** 1Department of Endocrinology and Metabolism, Chosun University Hospital, Chosun University School of Medicine, Gwangju 61453, Republic of Korea; lyu0923@chosun.ac.kr (Y.S.L.); endocrine@chosun.ac.kr (J.H.K.); 2Division of Nephrology, Department of Medicine, Chosun University Hospital, Chosun University School of Medicine, Gwangju 61453, Republic of Korea; korean8503@chosun.ac.kr

**Keywords:** periodontitis, body size phenotype, metabolic syndrome, KNHANES

## Abstract

We aimed to examine the correlation between periodontitis and body size phenotypes in 7301 participants without diagnosed chronic diseases in the Korean National Health and Nutrition Examination Survey 2013–2015. The participants were categorized into the following body size phenotype groups based on body mass index and the presence of metabolic syndrome: metabolically healthy normal weight (MHNW), metabolically abnormal normal weight (MANW), metabolically healthy obese (MHO), and metabolically abnormal obese (MAO). The prevalence rates of mild and severe periodontitis were 18.1% and 7.5%, respectively. Patients with periodontitis were older, current smokers, had a lower family income, were less likely to engage in regular tooth brushing or exercise, and had a higher body mass index and glucose levels. Periodontitis was more prevalent in the MANW and MAO groups than in the MHNW and MHO groups. Compared with the MHNW phenotype, the MAO and MANW phenotypes were significantly associated with mild and severe periodontitis, and the MHO phenotype was significantly associated with mild periodontitis. The MANW and MAO phenotypes are independent risk factors for periodontitis in adults without diagnosed chronic diseases. To enhance public health, a greater focus and effective approaches for identifying metabolic disease phenotypes among individuals with periodontal disease may be clinically relevant.

## 1. Introduction

Periodontitis, a common inflammatory condition affecting tissues surrounding the teeth, is characterized by the progressive destruction of the supporting structures of the teeth, including the gingiva, periodontal ligament, and alveolar bone [1]. This condition not only leads to tooth loss and oral health problems but is also widely recognized for its association with various systemic diseases, particularly metabolic syndrome [2].

Metabolic syndrome is defined as a cluster of conditions including hypertension, dyslipidemia, hyperglycemia, and abdominal obesity, all of which significantly increase the risk of cardiovascular diseases and diabetes [3]. The connection between periodontitis and metabolic syndrome is thought to be mediated by chronic systemic inflammation. Inflammatory mediators such as C-reactive protein (CRP), interleukin-6 (IL-6), and tumor necrosis factor-alpha (TNF-α) are elevated in both periodontitis and metabolic syndrome, suggesting a bidirectional relationship where each condition may exacerbate the other [2,4,5,6]. The presence of periodontitis may serve as a marker for systemic inflammatory burden, highlighting the importance of oral health in the management and prevention of broader systemic health issues [4,7,8]. Similarly, obesity, which is often a component of metabolic syndrome, is known to exacerbate periodontal disease due to an increased inflammatory burden and altered immune responses [9]. Adipose tissue, particularly in obese individuals, secretes proinflammatory cytokines that can contribute to the systemic inflammation observed in periodontitis [10].

The concept of body size phenotypes, which integrates both metabolic syndrome and obesity, is used to classify individuals into four distinct phenotype categories [11]: metabolically healthy normal weight (MHNW), metabolically abnormal normal weight (MANW), metabolically healthy obese (MHO), and metabolically abnormal obese (MAO). Recognizing this classification is clinically significant because it reveals the diverse cardiovascular risk profiles of each phenotype. Obesity phenotypes are linked to varying clinical traits, potentially leading to different cardiovascular outcomes and mortality risks [12,13]. This approach is essential because it highlights the fact that a normal body mass index (BMI) does not necessarily indicate metabolic health; similarly, not all individuals with obesity are metabolically unhealthy [14]. By identifying at-risk individuals based on their specific body size phenotype, healthcare providers can implement targeted interventions early to prevent adverse metabolic profiles and mitigate the risk of negative clinical outcomes in the future.

Considering the diverse clinical courses associated with each body size phenotype, personalized treatment approaches are essential. Current evidence has established analyses on the relationships between periodontitis and individual metabolic components, such as diabetes [6,15], metabolic syndrome [2], and obesity [9,16]. However, there is limited evidence on the relationship between periodontitis and body size phenotype. Therefore, the aim of this study was to investigate the association between periodontitis and various body size phenotypes in a cohort of seemingly healthy adults, using data from the Korean National Health and Nutrition Examination Survey (KNHANES) conducted between 2013 and 2015.

## 2. Materials and Methods

### 2.1. Study Population

Data from the Korean National Health and Nutrition Examination Survey (KNHANES) conducted by the Korean Ministry of Health and Welfare between 2013 and 2015 were used. Participants were selected using a stratified, multistage, and clustered probability sampling method based on household registry data including geographic location, sex, and age categories [17]. This cross-sectional and nationally representative study included non-institutionalized civilians and involved a health interview and nutrition and health examination conducted by trained investigators. All participants provided informed consent, and this study was approved by the Institutional Review Board of Chosun University Hospital (approval number: 2022-07-010).

Of the initial 22,948 participants, this study focused on 17,780 adults (7709 men and 10,071 women) aged > 20 years. Individuals with chronic diseases (e.g., hypertension, diabetes, dyslipidemia, cardiovascular diseases, cerebral infarction, chronic renal/liver failure, or malignancy), those who had fasted for less than 8 h, and those with missing data were excluded. A healthy population was defined by the absence of any diagnosed chronic disease. The final analysis included 7301 adults without diagnosed chronic diseases (3206 men and 4095 women).

### 2.2. Measurement and Classification of Variables

Height was determined using a portable stadiometer (Seriter, Bismarck, ND, USA), while body weight was measured with a balance scale (Giant-150N, Hana, Seoul, Republic of Korea). Body mass index (BMI) was calculated by dividing the weight in kilograms by the square of the height in meters. Waist circumference was taken at the midpoint between the lower rib margin and the iliac crest by trained personnel. Blood pressure (BP) was recorded using a mercury sphygmomanometer after the participant had rested for 5 min in a seated position; the average of two separate BP readings was used in the analysis. Venous blood samples were drawn following a fasting period of at least 8 h [18], and the levels of plasma glucose, high-density lipoprotein cholesterol (HDL-C), low-density lipoprotein cholesterol (LDL-C), and triglyceride (TG) were assessed with a Hitachi Automatic Analyzer 7600. Glycated hemoglobin (HbA1c) was measured using high-performance liquid chromatography (Tosoh G8). Additionally, self-reported questionnaires were employed to collect data on participants’ residential location, education level, household income, smoking habits, alcohol intake, regular physical activity, and total energy consumption.

#### 2.2.1. Body Size Phenotype

Participants were classified into four body size phenotypes based on BMI and the presence of metabolic syndrome: metabolically healthy normal weight (MHNW), metabolically abnormal normal weight (MANW), metabolically healthy obese (MHO), and metabolically abnormal obese (MAO). Metabolic syndrome was defined by the presence of at least three of the following criteria: abdominal obesity (waist circumference ≥ 90 cm for males or ≥85 cm for females, according to the Korean Society of Obesity), hypertriglyceridemia (serum TG levels ≥ 150 mg/dL or current treatment for this condition), low HDL-C (serum HDL-C levels < 40 mg/dL for males or <50 mg/dL for females, or ongoing treatment for this lipid abnormality), high blood pressure (systolic blood pressure ≥ 130 mmHg and diastolic blood pressure ≥ 85 mmHg, or use of antihypertensive medication), and high fasting blood glucose (fasting glucose levels ≥ 100 mg/dL or treatment with antidiabetic medication).

Participants were categorized as MHNW if they had a normal BMI (18.5–24.9 kg/m^2^) and no presence of metabolic syndrome. Those with a normal BMI but exhibiting metabolic syndrome were classified as MANW. Individuals with a BMI of 25 kg/m^2^ or higher and no presence of metabolic syndrome were classified as MHO, whereas those with a BMI of 25 kg/m^2^ or higher and with metabolic syndrome were categorized as MAO.

#### 2.2.2. Periodontitis

An oral health survey was conducted by calibrated dentists. Periodontal disease was assessed using the Community Periodontal Index (CPI), graded on a 4-point scale: 0 (healthy), 1 (gingival bleeding after probing), 2 (calculus), 3 (pocket probing depth [PPD] between 3.5 mm and 5.5 mm), and 4 (PPD > 5.5 mm). The highest score for each sextant was recorded. Periodontal status was categorized as severe periodontitis (CPI = 4), mild periodontitis (CPI = 3), or no periodontitis (CPI ≤ 2 in all sextants) [19,20].

### 2.3. Statistical Analysis

A complex sample analysis was performed on the KNHANES dataset following the guidelines set by the Korean Centers for Disease Control and Prevention. Continuous variables were expressed as means with standard deviations, while categorical variables were reported as weighted percentages. The chi-square test was used to evaluate differences in general characteristics between groups. To identify risk factors associated with periodontitis, multivariate logistic regression analyses were conducted, adjusting for variables such as age, sex, BMI, waist circumference, systolic blood pressure, diastolic blood pressure, TG, HDL-C, LDL-C, HbA1c, sociodemographic factors (including residential location, family income, and education level), and lifestyle behaviors (such as smoking, alcohol use, regular physical activity, and total energy intake). All statistical analyses were carried out using the Statistical Package for the Social Sciences (SPSS) software version 25.0 (IBM SPSS, Version 25.0. Armonk, NY, USA: IBM Corp), with p-values less than 0.05 deemed statistically significant.

## 3. Results

### 3.1. Basic Demographic and Clinical Characteristics

Table 1 shows the baseline characteristics of the study population according to periodontal status. The prevalence rates of mild and severe periodontitis were 18.1% and 7.5%, respectively. Patients with severe periodontitis tended to be older, had a higher BMI, were current smokers, had a lower family income, were less likely to engage in regular tooth brushing or exercise, and were more likely to live in rural areas. No significant difference in alcohol consumption was observed between the subgroups. Patient groups with severe periodontitis had higher proportions of individuals with the MANW and MAO phenotypes, compared with those without periodontitis.

### 3.2. Clinical Characteristics and Periodontal Status According to Body Size Phenotype

Table 2 shows the characteristics of the study population according to body size phenotype. The proportions of individuals with the MHNW, MANW, MHO, and MAO phenotypes were 67.9%, 4.9%, 16.3%, and 10.9%, respectively.

Individuals with severe periodontitis were more likely to fall into the MANW and MAO categories, indicating a higher prevalence of metabolically abnormalities, irrespective of BMI. These individuals were older and had higher FPG, HbA1c, and triglyceride levels. They also had lower HDL-C levels, were more likely to smoke, and were less likely to brush their teeth regularly or engage in regular exercise.

### 3.3. Association between Periodontal Status and Body Size Phenotype

Table 3 summarizes the outcomes of the logistic regression analysis examining the link between periodontitis and body size phenotype. In the analysis group categorized by the MAO and MHNW phenotypes, the MAO phenotype showed a significant association with both mild and severe periodontitis (odds ratio [OR] 1.489 [95% confidence interval (CI) 1.236–1.795] and OR 1.945 [1.527–2.478], respectively) compared to those without periodontitis. This association persisted even after adjusting for variables such as age, sex, sociodemographic factors (including place of residence, family income, and education), lifestyle behaviors (such as smoking status, alcohol consumption, regular physical activity, total energy intake, carbohydrate, protein, and fat intake), and dental factors (such as frequency of tooth brushing and regular dental checkups).

For the MANW and MHNW phenotypes, mild and severe periodontitis were significantly associated with the MANW phenotype (OR, 1.488 [1.167–1.898] and OR, 1.769 [1.291–2.425], respectively), compared with no periodontitis. When comparing the MHO and MHNW phenotypes, mild periodontitis was more significantly associated with the MHO phenotype (OR, 1.413 [1.166–1.711]), compared with the MHNW phenotype. However, severe periodontitis did not show a significant association with the MHO phenotype (OR, 1.281 [0.953–1.723]).

Comparisons between the MAO and MANW phenotypes and between the MANW and MHO phenotypes did not show significant associations for mild periodontitis. However, severe periodontitis was significantly associated with the MANW phenotype compared with the MHO phenotype (OR, 1.616 [1.118–2.335]). Finally, when comparing the MAO and MHO phenotypes, severe periodontitis was more significantly associated with the MAO phenotype (OR, 1.559 [1.141–2.129]), while mild periodontitis did not show a significant association (OR, 1.045 [0.821–1.330]).

## 4. Discussion

This study provides significant insights into the relationship between periodontitis and body size phenotypes, emphasizing the relationship between oral health and systemic metabolic conditions. These findings revealed that severe periodontitis was associated with adverse metabolic phenotypes. This highlights the need for integrated health approaches that address both periodontal and metabolic health.

This study revealed that the prevalence of periodontitis increased across the MAO, MANW, and MHO phenotype groups, compared with the MANW phenotype group. This association suggests that periodontitis may serve as a marker for metabolic syndrome and other systemic inflammatory conditions, thereby providing a valuable opportunity for early intervention. These findings are consistent with the results from several other studies, which revealed increased prevalence rates of periodontitis in patients with diabetes [15,21,22], metabolic syndrome [2], and obesity [9,16]. The recognition of periodontitis as a potential indicator of broader systemic issues reinforces the importance of routine periodontal assessments as part of comprehensive health evaluations, particularly for those in high-risk phenotypic categories, particularly those in the MAO and MANW phenotype groups.

Interestingly, our study found that the MANW phenotype was more strongly associated with severe periodontitis compared to the MHO phenotype. The underlying mechanisms that could account for the varying risk of periodontitis in the MANW or MHO phenotypes are not well understood. However, it is notable that some studies suggested that individuals with the MANW phenotype may have worse clinical outcomes than those with the MHO phenotype or those in the general population. For example, a prospective study from Korea reported that the MANW phenotype showed increased arterial stiffness and carotid intima–media thickness compared to the MANW phenotype [23]. Similarly, another prospective study from Korea reported that the MANW phenotype had an unfavorable inflammatory and atherogenic profile, including higher levels of inflammatory markers, smaller LDL particles, and oxidized LDL compared to the MHO phenotype [24]. In contrast, individuals with the MHO phenotype may have higher adiponectin [25] and lower levels of systemic inflammation [26] compared to those with the MAO phenotype, leading to a weaker association with severe periodontitis. This could be due to the higher levels of systemic inflammation in MANW individuals which may contribute to their increased susceptibility to severe periodontitis [27]. In contrast, individuals with the MHO phenotype, who have obesity without metabolic syndrome, may have higher adiponectin [25] and lower levels of systemic inflammation [26] compared to those with the MAO phenotype, leading to a weaker association with severe periodontitis. Individuals with the MANW phenotype often appeared non-obese, leading to delayed diagnosis. However, this group had a high prevalence of diabetes and poor long-term prognosis due to delayed diagnosis [28,29]. Screening for metabolic syndrome in patients with periodontitis, even if they are believed to have no underlying conditions, could aid in the early detection of the MANW phenotype.

Importantly, this study found that periodontitis was more prevalent in the MHO phenotype group than in the MANW phenotype group. While several studies have concluded that the MHO phenotype is metabolically stable and, therefore, a benign entity [30,31], compared with the MAO phenotype, our findings indicate an increased risk of periodontitis in this group compared with the MHNW phenotype group. This suggests that individuals with the MHO phenotype require careful monitoring and evaluation. Consequently, it is crucial to pay close attention to those with the MHO phenotype as well, ensuring that they receive appropriate surveillance and intervention to effectively manage potential chronic inflammation.

In this study, it was observed that individuals with severe periodontitis tend to exercise less regularly, have higher energy intake, and intake more carbohydrates compared to those without periodontitis (Table 1). These lifestyle habits can increase the risk of metabolic syndrome, including components such as obesity and diabetes [32]. The increase in metabolic syndrome, in turn, can exacerbate systemic inflammation, which may further worsen periodontitis. Several studies have also confirmed that the incidence of periodontitis is higher in individuals who exercise less [33,34] and intake more carbohydrates [35]. Based on this evidence, it can be concluded that adopting a healthy lifestyle not only improves metabolic syndrome but may also help in managing periodontitis.

Owing to the cross-sectional nature of our study, we could not establish a causal link between periodontitis and metabolic phenotypes. However, there are several possible explanations for this finding. The relationship between periodontitis and metabolic phenotypes may be attributed to alterations in the oral microbiome. The human mouth hosts a diverse population of microorganisms, including bacteria, viruses, and fungi, collectively known as the oral microbiome [36]. This microbiome is essential for maintaining oral health, and any imbalance can lead to poor metabolic control and increased inflammation. Previous studies have shown structural changes in the oral microbiome of patients with obesity compared with healthy controls [37,38,39]. Additionally, oral dysbiosis and periodontal disease are associated with systemic inflammation, which contributes to the worsening of metabolically abnormalities, such as diabetes [36,39] and obesity [40,41]. Collectively, there is strong evidence that oral bacteria act as upstream triggers of adipose tissue inflammation and subsequent metabolic diseases.

Another possible mechanism is that periodontitis, as a complex, chronic inflammatory condition, can lead to the worsening of the metabolic phenotype. This inflammatory response is marked by an abnormal secretion of host-derived factors that contribute to inflammation and tissue degradation. Periodontitis and adverse metabolic phenotypes share the underlying mechanism of chronic inflammatory diseases. Metabolically, abnormalities induce various proinflammatory effects, including the release of TNF-α, IL-6, and leptin, which affect multiple body systems, including the oral cavity [21]. D’Aiuto et al. demonstrated that chronic inflammation, a hallmark of metabolic disorders, can initiate processes leading to reduced macrophage and neutrophil functions, advanced glycosylation accumulation, and inflammation, all of which contribute to the development of periodontitis [42]. Additionally, periodontitis and adverse metabolic phenotypes share common risk factors such as lower income, low education levels, less exercise, less frequent tooth brushing, and high carbohydrate intake. These shared poor lifestyle habits are likely to be risk factors contributing to the development of both periodontitis and metabolic phenotypes.

The strength of our study lies in the use of an extensive and nationally representative KNHANES dataset, which covers clinical, sociodemographic, lifestyle, and anthropometric data. Furthermore, to improve the reliability of this study, we focused on patients without any known underlying conditions, aiming to reduce potential confounding factors. However, this study has several limitations that should be acknowledged. First, the cross-sectional design limited our ability to establish causality between periodontitis and metabolic phenotypes. Longitudinal studies are needed to better understand the temporal relationships and potential causal pathways. Second, the reliance on self-reported data for some variables, such as smoking status and physical activity, may have introduced reporting bias. Despite these limitations, this study’s large sample size and the use of nationally representative data enhanced the generalizability of the findings. Third, there is a possibility of the dataset containing unaccounted for and residual confounding variables, which are prevalent issues in observational studies. Fourth, although self-reported questionnaires are efficient and cost-effective, they can be limited by variations in health awareness and differences among groups, which might lead to biased results. Last, we did not separate and analyze the results by sex, which could have provided insights into potential differences between males and females in relation to periodontitis, metabolic syndrome components, and lifestyle factors. Future studies should consider sex-specific analyses to better understand these differences and improve the applicability of the findings.

## 5. Conclusions

This study highlights the significant association between periodontitis and adverse metabolic phenotypes, particularly the MANW and MAO categories, in adults without diagnosed chronic diseases. Screening for metabolic syndrome in patients with periodontitis, even if they are believed to have no underlying conditions, could aid in the early detection of metabolic syndrome.

## Figures and Tables

**Table 1 ijerph-21-01180-t001:** The characteristics of the study population according to periodontal status.

	No Periodontitis	Mild Periodontitis	Severe Periodontitis	*p*-Value
Number (%)	5436 (74.5)	1318 (18.1)	547 (7.5)	
Sex (male, %)	40.0	52.7	62.0	0.000
Age (years)	38.51 ± 0.22	48.03 ± 0.42	52.23 ± 0.47	0.000
BMI (kg/m^2^)	23.26 ± 0.06	24.10 ± 0.10	24.13 ± 0.13	0.000
WC (cm)	79.13 ± 0.17	82.60 ± 0.29	83.18 ± 0.38	0.000
SBP (mmHg)	112.1 ± 0.23	117.51 ± 0.48	118.83 ± 0.72	0.000
DBP (mmHg)	74.05 ± 0.19	77.12 ± 0.34	77.44 ± 0.45	0.000
FPG (mg/dL)	93.40 ± 0.21	99.18 ± 0.63	101.31 ± 0.97	0.000
HbA1c (%)	5.51 ± 0.01	5.70 ± 0.02	5.83 ± 0.03	0.000
LDL-C (mg/dL)	114.13 ± 0.71	118.87 ± 1.26	119.77 ± 2.48	0.001
HDL-C (mg/dL)	52.68 ± 0.18	49.02 ± 0.35	49.00 ± 0.53	0.000
TG (mg/dL)	122.1 ± 1.55	153.79 ± 3.99	154.86 ± 5.95	0.000
Family income percentile (%)				0.000
<25	9.0	12.0	14.4	
25–50	23.5	27.0	25.0	
50–75	32.4	31.8	29.5	
≥75	35.2	29.1	31.0	
More than high school education (%)	88.9	76.7	67.9	0.000
Residence in urban area (%)	85.8	79.5	73.5	0.000
Smoking				0.000
Never	58.4	42.4	33.7	
Past	17.5	19.7	24.3	
Current	21.7	34.4	37.3	
Alcohol drinking (yes, %)	62.9	61.9	64.3	0.630
Regular exercise ^a^ (yes, %)	59.3	53.5	46.6	0.000
Total energy intake (kcal)	2174.90 ± 16.42	2206.38 ± 33.73	2220.22 ± 56.89	0.001
Carbohydrate intake (g)	314.71 ± 2.16	327.65 ± 4.39	335.29 ± 6.74	0.000
Protein intake (g)	77.04 ± 0.72	76.69 ± 1.48	76.50 ± 3.96	0.964
Fat intake (g)	53.22 ± 0.66	47.31 ± 1.23	45.75 ± 2.33	0.000
Tooth brushing ≥ 3 times/day (%)	57.4	48.6	46.0	0.000
Periodic dental checkup (%) ^b^	30.7	29.7	29.7	0.719
Metabolic phenotype				0.000
MHNW	70.6	56.7	54.8	
MANW	3.5	7.2	11.4	
MHO	16.8	18.9	14.5	
MAO	9.1	17.2	19.4	

Data are expressed as the mean ± SE for continuous variables and as weighted percentages for categorical variables. BMI, body mass index; DBP, diastolic blood pressure; FPG, fasting plasma glucose; MANW, metabolically abnormal normal weight; MAO, metabolically abnormal obese; MHNW, metabolic healthy normal weight; MHO, metabolic healthy obese; SBP, systolic blood pressure; TG, triglyceride; WC, waist circumference. ^a^ Regular exercise was defined as engaging in physical activity for at least 2.5 h of moderate-intensity physical activity per week or 1.25 h of high-intensity physical activity. ^b^ A periodic dental checkup was defined as a case of having a dental checkup at least once during the past year.

**Table 2 ijerph-21-01180-t002:** The characteristics of the study population according to body size phenotypes.

	MHNW	MANW	MHO	MAO	*p*-Value
Number (%)	4959 (67.9)	359 (4.9)	1190 (16.3)	793 (10.9)	
Sex (male, %)	38.6	54.9	51.0	61.4	0.000
Age (years)	38.78 ± 0.24	57.12 ± 0.52	42.05 ± 0.36	50.66 ± 0.37	0.000
BMI (kg/m^2^)	21.45 ± 0.03	23.08 ± 0.05	27.21 ± 0.05	28.43 ± 0.07	0.000
WC (cm)	74.59 ± 0.12	83.04 ± 0.23	88.33 ± 0.19	94.12 ± 0.20	0.000
SBP (mmHg)	111.18 ± 0.19	128.49 ± 0.59	117.03 ± 0.31	127.62 ± 0.36	0.000
DBP (mmHg)	71.60 ± 0.14	79.08 ± 0.39	75.64 ± 0.24	81.92 ± 0.29	0.000
FPG (mg/dL)	93.46 ± 0.20	117.05 ± 1.18	96.34 ± 0.34	113.47 ± 0.71	0.000
HbA1c (%)	5.54 ± 0.01	6.30 ± 0.04	5.65 ± 0.01	6.16 ± 0.03	0.000
LDL-C (mg/dL)	108.61 ± 0.60	111.39 ± 1.45	117.95 ± 1.06	117.60 ± 1.03	0.000
HDL-C (mg/dL)	54.56 ± 0.15	41.77 ± 0.30	50.33 ± 0.24	41.99 ± 0.22	0.000
TG (mg/dL)	100.23 ± 0.87	239.87 ± 5.68	125.78 ± 2.08	235.68 ± 4.48	0.000
Family income percentile (%) ^a^					0.000
<25	9.0	17.2	10.0	11.4	
25–50	24.2	25.5	25.0	24.1	
50–75	32.4	27.4	31.8	35.2	
≥75	34.5	29.9	33.1	29.4	
More than high school education (%)	87.4	67.7	85.9	79.6	0.000
Residence in urban area (%)	85.6	78.7	83.7	80.1	0.000
Smoking					0.000
Never	58.8	39.1	47.9	37.8	
Past	16.6	20.1	18.3	22.6	
Current	22.1	36.2	30.2	35.0	
Alcohol drinking (yes, %)	62.2	63.8	63.3	67.6	0.041
Regular exercise ^a^ (yes, %)	57.3	54.0	63.1	51.9	0.003
Total energy intake (kcal)	2115.97 ± 13.70	1999.18 ± 36.87	2184.69 ± 26.61	2150.62 ± 28.20	0.000
Carbohydrate intake (g)	316.56 ± 1.90	314.92 ± 4.57	323.32 ± 3.50	323.47 ± 3.53	0.068
Protein intake (g)	74.27 ± 0.66	66.80 ± 1.50	77.46 ± 1.21	73.92 ± 1.21	0.000
Fat intake (g)	50.49 ± 0.53	36.76 ± 1.26	51.45 ± 1.07	44.36 ± 1.08	0.000
Tooth brushing ≥ 3 times/day (%)	58.2	44.6	50.8	45.1	0.000
Periodic dental checkup (%) ^b^	31.6	26.7	31.4	25.3	0.044
Periodontitis					0.000
None	79.7	56.7	74.8	61.5	
Mild periodontitis	14.5	26.2	19.1	26.2	
Severe periodontitis	5.8	17.1	6.1	12.3	

Data are expressed as the mean ± SE for continuous variables and as weighted percentages for categorical variables. BMI, body mass index; DBP, diastolic blood pressure; FPG, fasting plasma glucose; MANW, metabolically abnormal normal weight; MAO, metabolically abnormal obese; MHNW, metabolic healthy normal weight; MHO, metabolic healthy obese; SBP, systolic blood pressure; TG, triglyceride; WC, waist circumference. ^a^ Regular exercise was defined as engaging in physical activity for at least 2.5 h of moderate-intensity physical activity per week or 1.25 h of high-intensity physical activity. ^b^ A periodic dental checkup was defined as a case of having a dental checkup at least once during the past year.

**Table 3 ijerph-21-01180-t003:** Association between periodontitis and body size phenotypes.

	Fully Adjusted OR (95% Cl)				
	MAO		MANW		MHO
MAO/MHNW		MANW/MHNW		MHO/MHNW	
No periodontitis	Reference	No periodontitis	Reference	No periodontitis	Reference
Mild periodontitis	1.489 (1.236–1.795)	Mild periodontitis	1.488 (1.167–1.898)	Mild periodontitis	1.413 (1.166–1.711)
Severe periodontitis	1.945 (1.527–2.478)	Severe periodontitis	1.769 (1.291–2.425)	Severe periodontitis	1.281 (0.953–1.723)
MAO/MANW		MANW/MHO			
No periodontitis	Reference	No periodontitis	Reference		
Mild periodontitis	0.983 (0.754–1.283)	Mild periodontitis	1.115 (0.821–1.514)		
Severe periodontitis	1.034 (0.747–1.432)	Severe periodontitis	1.616 (1.118–2.335)		
MAO/MHO					
No periodontitis	Reference				
Mild periodontitis	1.045 (0.821–1.330)				
Severe periodontitis	1.559 (1.141–2.129)				

CI, confidence interval; MANW, metabolically abnormal normal weight; MAO, metabolically abnormal obese; MHNW, metabolic healthy normal weight; MHO, metabolic healthy obese; OR, odds ratio. Logistic models are adjusted for age, sex, sociodemographic factors (place of residence, family income, and education), lifestyle behaviors (smoking status, alcohol consumption, regular exercise, total energy intake, carbohydrate intake, protein intake, and fat intake), and dental factors (number of tooth brushing and periodic dental checkup).

## Data Availability

Information can be accessed from the Korea National Health and Nutrition Examination Survey (KNHANES), which is organized by the Korea Centers for Disease Control and Prevention (KCDCP). The data are freely available on the KCDCP website (https://knhanes.cdc.go.kr, accessed on 21 July 2024).

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
