# Peer review of "The Effect of Periodontitis on Body Size Phenotypes in Adults without Diagnosed Chronic Diseases: The Korean National Health and Nutrition Examination Survey 2013–2015"

_ijerph, 2024, doi:10.3390/ijerph21091180_

Round 1

Reviewer 1 Report

Comments and Suggestions for Authors

The paper aims to investigate the relationship between periodontitis and body size phenotypes among participants in the Korean National Health and Nutrition Examination Survey conducted from 2013 to 2015. It thoroughly explains the concept of body size phenotypes, which encompass both metabolic syndrome and obesity. Notably, it highlights that a normal body mass index (BMI) does not necessarily equate to metabolic health, just as not all individuals with obesity are metabolically unhealthy. Thank you authors for this study.

Abstract: examine the correlation between periodontitis and body size phenotypes in 7,301 apparently healthy participants of the Korean National Health and Nutrition Examination Survey 2013–2015.

Line 13 It is not clear what you mean by ''apparently healthy participants''- please rephrase. Especially as metabolic syndrome and obesity is mentioned below- so it is not so correct to call them 'healthy patients'.

This is also in the title-which could be improved.

M&M:

2.2 Measurements and classification

It is not clear to me how MHNW, MANW, MHO and MAO are categorized, could you expand and clarify this better in respect to the blood tests and variables.

Line 102: Please double check your categories for CPI codes and severity of periodontitis. 

Results:

Since you have the data for both male and female, results which are also adjusted to sex and a large enough sample, could you distinguish and compare the both? It is important to know if there is a difference in results when this is separated, as there is a difference in periodontitis, a significant difference between males and females in metabolic syndrome components, in lifestyle factors, dental awareness etc. If not, please mention this in the discussion part the sex and gender differences and the limitations of your results in this respect.

Discussion:

Could you discuss something about your results/associations for individuals with severe periodontitis, metabolic syndrome and lifestyle from Lines 132-136?

As mentioned, the sex and gender part of your results and limitations, as commented above.

Author Response

Journal Revision Letter (Response to Reviewers)

Aug 31 2024

Response to Editor and Reviewers:

Thank you for the insightful and thorough review of our manuscript and your comments and suggestions. It is with excitement that I re-submit to you our revised version of manuscript entitled “Effect of periodontitis on body size phenotypes in apparently healthy adults: the Korean National Health and Nutrition Ex-amination Survey 2013-2015” for the IJERPH. We have revised our paper, taking your comments and suggestions into account. In the process, we believe that this paper has been significantly improved.

             We have responded specifically to each suggestion below.

Sincerely,

Sang Yong Kim, MD, PhD:

Department of Endocrinology and Metabolism, Chosun University Hospital,

588 Seoseok-Dong, Dong-Gu, Gwangju, Republic of Korea.

e-mail: diabetes@chosun.ac.kr

Office: 82-62-220-3011

ORCID: 0000-0002-3902-622X

Reviewer #1

The paper aims to investigate the relationship between periodontitis and body size phenotypes among participants in the Korean National Health and Nutrition Examination Survey conducted from 2013 to 2015. It thoroughly explains the concept of body size phenotypes, which encompass both metabolic syndrome and obesity. Notably, it highlights that a normal body mass index (BMI) does not necessarily equate to metabolic health, just as not all individuals with obesity are metabolically unhealthy. Thank you authors for this study.

Abstract: examine the correlation between periodontitis and body size phenotypes in 7,301 apparently healthy participants of the Korean National Health and Nutrition Examination Survey 2013–2015.

Line 13 It is not clear what you mean by ''apparently healthy participants''- please rephrase. Especially as metabolic syndrome and obesity is mentioned below- so it is not so correct to call them 'healthy patients'.

This is also in the title-which could be improved.

Reply) We appreciate the reviewer’s suggestion to clarify the terminology used to describe our study population. As you (Reviewer #1) point out, we have made the following revisions:

Original Title: "Effect of periodontitis on body size phenotypes in apparently healthy adults: the Korean National Health and Nutrition Examination Survey 2013-2015"

Revised Title: "Impact of Periodontitis on Body Size Phenotypes in Adults Without Diagnosed Chronic Diseases: Insights from the Korean National Health and Nutrition Examination Survey 2013-2015"

This revised title more accurately reflects the study population, emphasizing that while the participants are free of diagnosed chronic diseases, they may still exhibit metabolic abnormalities. Additionally, we have revised the term 'apparently healthy adults' throughout the manuscript as follows.

"We aimed to examine the correlation between periodontitis and body size phenotypes in 7,301 participants without diagnosed chronic diseases of the Korean National Health and Nutrition"

"The MANW and MAO phenotypes are independent risk factors for periodontitis in adults without diagnosed chronic diseases."

"The final analysis included 7,301 adults without diagnosed chronic diseases (3,218 men and 4,417 women)."

"This study highlights the significant association between periodontitis and adverse metabolic phenotypes, particularly the MANW and MAO categories, in adults without diagnosed chronic diseases."

2.2 Measurements and classification

It is not clear to me how MHNW, MANW, MHO and MAO are categorized, could you expand and clarify this better in respect to the blood tests and variables.

Reply) Thank you for pointing out this important issue. We totally agree that further clarification is necessary regarding the categorization of the body size phenotypes (MHNW, MANW, MHO, MAO). In response, we have created a new section, 2.2.1, and revised it as follows:

“2.2.1. Body size phenotype

Participants were classified into four body size phenotypes based on BMI and the presence of metabolic syndrome: Metabolically Healthy Normal Weight (MHNW), Metabolically Abnormal Normal Weight (MANW), Metabolically Healthy Obese (MHO), and Metabolically Abnormal Obese (MAO). Metabolic syndrome was defined by the presence of at least three of the following criteria: abdominal obesity (waist cir-cumference ≥90 cm for males or ≥85 cm for females, according to the Korean Society of Obesity), hypertriglyceridemia (serum TG levels ≥150 mg/dL or current treatment for this condition), low HDL-C (serum HDL-C levels <40 mg/dL for males or <50 mg/dL for females, or ongoing treatment for this lipid abnormality), high blood pressure (systolic blood pressure ≥130 mmHg and diastolic blood pressure ≥85 mmHg, or use of antihy-pertensive medication), and high fasting blood glucose (fasting glucose levels ≥100 mg/dL or treatment with antidiabetic medication).

Participants were categorized as MHNW if they had a normal BMI (18.5–24.9 kg/m²) and no presence of metabolic syndrome. Those with a normal BMI but exhibit-ing metabolic syndrome were classified as MANW. Individuals with a BMI of 25 kg/m² or higher and no presence of metabolic syndrome were classified as MHO, whereas those with a BMI of 25 kg/m² or higher and with metabolic syndrome were categorized as MAO.”

Line 102: Please double check your categories for CPI codes and severity of periodontitis.

Reply) Thank you for pointing my manuscript error. As you (Reviewer #1) point out, we have made the following revisions:

“Periodontal status was categorized as severe periodontitis (CPI = 4), mild periodontitis (CPI = 3), or no periodontitis (CPI ≤ 2 in all sextants).”

Results:

Since you have the data for both male and female, results which are also adjusted to sex and a large enough sample, could you distinguish and compare the both? It is important to know if there is a difference in results when this is separated, as there is a difference in periodontitis, a significant difference between males and females in metabolic syndrome components, in lifestyle factors, dental awareness etc. If not, please mention this in the discussion part the sex and gender differences and the limitations of your results in this respect.

Reply) Thank you for your insightful suggestion. We recognize the importance of distinguishing and comparing results between males and females, especially given the known differences in periodontitis, metabolic syndrome components, lifestyle factors, and dental awareness between the sexes. However, due to the scope of this study and the current analysis limitation, we did not separate the results by sex. We agree that this represents a limitation of our study, and we will address this in the Discussion section. Specifically, we will mention that future research should consider sex and gender differences to provide a more nuanced understanding of the associations we have identified. We have added this as a limitation in the discussion as follows.

“Last, we did not separate and analyze the results by sex, which could have provided insights into potential differences between males and females in relation to periodon-titis, metabolic syndrome components, and lifestyle factors. Future studies should consider sex-specific analyses to better understand these differences and improve the applicability of the findings.”

Discussion:

Could you discuss something about your results/associations for individuals with severe periodontitis, metabolic syndrome and lifestyle from Lines 132-136?

Reply) Thank you for pointing out this important issue. As you pointed out, we have added the following paragraph to the discussion section.

“In this study, it was observed that individuals with severe periodontitis tend to exercise less regularly, have higher energy intake, and intake more carbohydrates compared to those without periodontitis (table 1). These lifestyle habits can increase the risk of metabolic syndrome, such as obesity and diabetes [32]. The increase in met-abolic syndrome, in turn, can exacerbate systemic inflammation, which may further worsen periodontitis. Several studies have also confirmed that the incidence of perio-dontitis is higher in individuals who exercise less [33,34] and intake more carbohy-drates [35]. Based on this evidence, it can be concluded that adopting a healthy lifestyle not only improves metabolic syndrome but may also help in managing periodontitis.”

As mentioned, the sex and gender part of your results and limitations, as commented above.

Reply) As you pointed out above mention, we added this as a limitation in the discussion as follows.

“Last, we did not separate and analyze the results by sex, which could have provided insights into potential differences between males and females in relation to periodon-titis, metabolic syndrome components, and lifestyle factors. Future studies should consider sex-specific analyses to better understand these differences and improve the applicability of the findings.”

Reviewer 2 Report

Comments and Suggestions for Authors

In the conclusion section of the article, the authors write just one sentence directly connected with the results of the study, while the other two sentences are common knowledge not extracted from the essence of the article. Statements like "Screening for metabolic syndrome in patients with periodontitis, even if they are believed to have no underlying conditions, could aid in the early detection of metabolic syndrome" would be more appropriate and targeted for this particular study.

Author Response

Journal Revision Letter (Response to Reviewers)

Aug 31 2024

Response to Editor and Reviewers:

Thank you for the insightful and thorough review of our manuscript and your comments and suggestions. It is with excitement that I re-submit to you our revised version of manuscript entitled “Effect of periodontitis on body size phenotypes in apparently healthy adults: the Korean National Health and Nutrition Ex-amination Survey 2013-2015” for the IJERPH. We have revised our paper, taking your comments and suggestions into account. In the process, we believe that this paper has been significantly improved.

             We have responded specifically to each suggestion below.

Sincerely,

Sang Yong Kim, MD, PhD:

Department of Endocrinology and Metabolism, Chosun University Hospital,

588 Seoseok-Dong, Dong-Gu, Gwangju, Republic of Korea.

e-mail: diabetes@chosun.ac.kr

Office: 82-62-220-3011

ORCID: 0000-0002-3902-622X

Reviewer #2

In the conclusion section of the article, the authors write just one sentence directly connected with the results of the study, while the other two sentences are common knowledge not extracted from the essence of the article. Statements like "Screening for metabolic syndrome in patients with periodontitis, even if they are believed to have no underlying conditions, could aid in the early detection of metabolic syndrome" would be more appropriate and targeted for this particular study.

Reply) Thank you for pointing out this important issue. We appreciate your suggestion to make the conclusion section more directly connected to the study's findings. We agree that the conclusion should emphasize the unique contributions of our research and provide actionable insights based on the results. Therefore, we will revise the conclusion to include targeted statements.

“This study highlights the significant association between periodontitis and adverse metabolic phenotypes, particularly the MANW and MAO categories, in adults without diagnosed chronic diseases. Screening for metabolic syndrome in patients with periodontitis, even if they are believed to have no underlying conditions, could aid in the early detection of metabolic syndrome.”

Reviewer 3 Report

Comments and Suggestions for Authors

Dear all,

The manuscript meets the IJERPH goal and provides useful information for scholars and professionals. The manuscript's extensiveness shows a strong awareness of Global Health issues. The insights presented have the potential to change the field and add to the literature. I appreciated the clear, well-written, and fascinating manuscript. However, some points are listed below:

Title

No comment

Abstract

No comment

Introduction

Could you provide references to explain why previous studies haven't evaluated the relationship between periodontitis and phenotype, as well as its specific risk factors?

Could you please include a brief section explaining the relationship between periodontal disease and systemic inflammatory burden?

Materials and Methods

Lines 76–77 state, "The final analysis included 7,301 individuals (3,218 men and 4,417 women)" Please check that adding the two numbers of males and females results in 7635, not 7301.

Line 88: ‘Self-reported questionnaires were used’: while self-reporting data offers speed and cost-effectiveness, its limitations stem from health awareness and group characteristics, potentially skewing the results.

Lines 111–112: ‘family history of diabetes’: the related data not found in the manuscript.

I would like to inquire about the participant's history of medical conditions. Have they taken any medications for other health conditions, such as anti-inflammatory drugs?

Line 78: In 2.2. Measurement and Classification of Variables: the portable stadiometer and balance scale are needed to notify the brand, model, and manufacture.

Lines 82–82: Waist circumference, blood pressure, and venous blood sample measurement methods need references.

Lines 98–103 need references to be supported.

Results

No comment

Discussion

One topic might be stated and added to the discussion section: which common risk factor and its mechanisms might be responsible for the acquired findings?

Please, could you deliberate on the values and significant impact of the current study in order to help resolve oral health issues and systematic metabolic conditions for specific population groups?

References

This section lacks sufficient references to support some of the ideas in the introduction and materials and methods sections.

Journal names should be abbreviated. Please follow the ACS style guide; it is recommended for IJERPH.

Best regards,

Author Response

Journal Revision Letter (Response to Reviewers)

Aug 31 2024

Response to Editor and Reviewers:

Thank you for the insightful and thorough review of our manuscript and your comments and suggestions. It is with excitement that I re-submit to you our revised version of manuscript entitled “Effect of periodontitis on body size phenotypes in apparently healthy adults: the Korean National Health and Nutrition Ex-amination Survey 2013-2015” for the IJERPH. We have revised our paper, taking your comments and suggestions into account. In the process, we believe that this paper has been significantly improved.

             We have responded specifically to each suggestion below.

Sincerely,

Sang Yong Kim, MD, PhD:

Department of Endocrinology and Metabolism, Chosun University Hospital,

588 Seoseok-Dong, Dong-Gu, Gwangju, Republic of Korea.

e-mail: diabetes@chosun.ac.kr

Office: 82-62-220-3011

ORCID: 0000-0002-3902-622X

Reviewer #3

The manuscript meets the IJERPH goal and provides useful information for scholars and professionals. The manuscript's extensiveness shows a strong awareness of Global Health issues. The insights presented have the potential to change the field and add to the literature. I appreciated the clear, well-written, and fascinating manuscript. However, some points are listed below:

Title

No comment

Abstract

No comment

Introduction

Could you provide references to explain why previous studies haven't evaluated the relationship between periodontitis and phenotype, as well as its specific risk factors?

Reply) Thank you for pointing out this important issue. Current evidence has established analyses on the relationships between periodontitis and individual metabolic components, such as diabetes, metabolic syndrome, and obesity. However, there has been limieted evidence on the relationship between periodontitis and body size phenotype. Therefore, this study was conducted to address this gap. To elaborate further, we have added additional references to current evidence on the relationship between periodontitis and metabolic syndrome on the introduction section as follows.

“Considering the diverse clinical courses associated with each body size phenotype, personalized treatment approaches are essential. Current evidence has established analyses on the relationships between periodontitis and individual metabolic components, such as diabetes, metabolic syndrome, and obesity. However, there has been limited evidence on the relationship between periodontitis and body size phenotype. Therefore, the aim of this study was to investigate the association be-tween periodontitis and various body size phenotypes in a cohort of seemingly healthy adults, using data from the Korean National Health and Nutrition Examination Survey (KNHANES) conducted between 2013 and 2015.”

Could you please include a brief section explaining the relationship between periodontal disease and systemic inflammatory burden?

Reply) Thank you for pointing out this important issue. As you correctly noted, our current manuscript lacked sufficient detail on the relationship between periodontal disease and systemic inflammatory burden. We revised the discussion section to include a explanation of the relationship between periodontal disease and systemic inflammatory burden.

“Owing to the cross-sectional nature of our study, we could not establish a causal link between periodontitis and metabolic phenotypes. However, there are several pos-sible explanations for this finding. The relationship between periodontitis and meta-bolic phenotypes may be attributed to alterations in the oral microbiome. The human mouth hosts a diverse population of microorganisms, including bacteria, viruses, and fungi, collectively known as the oral microbiome[36]. This microbiome is essential for maintaining oral health, and any imbalance can lead to poor metabolic control and in-creased inflammation. Previous studies have shown structural changes in the oral mi-crobiome of patients with obesity compared with healthy controls [37-39]. Additionally, oral dysbiosis and periodontal disease are associated with systemic inflammation, which contributes to the worsening of metabolically abnormalities, such as diabetes [36,39] and obesity [40,41]. Collectively, there is strong evidence that oral bacteria act as upstream triggers of adipose tissue inflammation and subsequent metabolic diseases.

Another possible mechanism is that periodontitis, as a complex, chronic inflam-matory condition, can lead to a worsening of the metabolic phenotype. This inflam-matory response is marked by an abnormal secretion of host-derived factors that con-tribute to inflammation and tissue degradation. Periodontitis and adverse metabolic phenotypes share the underlying mechanism of chronic inflammatory diseases. meta-bolically abnormalities induce various proinflammatory effects, including the release of TNF-α, IL-6, and leptin, which affect multiple body systems, including the oral cav-ity [21]. D'Aiuto et al. demonstrated that chronic inflammation, a hallmark of metabol-ic disorders, can initiate processes leading to reduced macrophage and neutrophil functions, advanced glycosylation accumulation, and inflammation, all of which con-tribute to the development of periodontitis [42]. Additionally, periodontitis and ad-verse metabolic phenotypes share common risk factors such as lower income, low ed-ucation levels, less exercise, less frequent tooth brushing, and high carbohydrate intake. These shared poor lifestyle habits are likely to be risk factors contributing to the de-velopment of both periodontitis and metabolic phenotypes.”

Materials and Methods

Lines 76–77 state, "The final analysis included 7,301 individuals (3,218 men and 4,417 women)" Please check that adding the two numbers of males and females results in 7635, not 7301.

Reply) Thank you for pointing my manuscript error. As you point out, we have made the following revisions.

“The final analysis included 7,301 adults without diagnosed chronic diseases (3,206 men and 4,095 women).”

Line 88: ‘Self-reported questionnaires were used’: while self-reporting data offers speed and cost-effectiveness, its limitations stem from health awareness and group characteristics, potentially skewing the results.

Reply) Thank you for pointing out this important issue. As you noted, this is one of the significant limitations of the study. The limitation has been added as follows

“Although self-reported questionnaires are efficient and cost-effective, they can be limited by variations in health awareness and differences among groups, which might lead to bi-ased results."

Lines 111–112: ‘family history of diabetes’: the related data not found in the manuscript.

Reply) Thank you for pointing my manuscript error. As you point out, we have made the following revisions.

“To identify risk factors associated with periodontitis, multivariate logistic regression analyses were conducted, adjusting for variables such as age, sex, BMI, waist circum-ference, systolic blood pressure, diastolic blood pressure, TG, HDL-C, LDL-C, HbA1c, sociodemographic factors (including residential location, family income, and education level), and lifestyle behaviors (such as smoking, alcohol use, regular physical activity, and total energy intake).”

I would like to inquire about the participant's history of medical conditions. Have they taken any medications for other health conditions, such as anti-inflammatory drugs?

Reply) Thank you for your insightful and important question. Unfortunately, we do not have information on medications such as anti-inflammatory drugs, so we were unable to include an analysis of this aspect.

Line 78: In 2.2. Measurement and Classification of Variables: the portable stadiometer and balance scale are needed to notify the brand, model, and manufacture.

Reply) Thank you for your comment. As you pointed out, I have revised the Methods as follows.

“Height was determined using a portable stadiometer (Seriter, Bismarck, ND, USA), while body weight was measured with a balance scale (Giant-150N, Hana, Seoul, Korea).”

Lines 82–82: Waist circumference, blood pressure, and venous blood sample measurement methods need references.

Reply) Thank you for your comment. As you pointed out, I have added the reference in the Methods section.

Lines 98–103 need references to be supported.

Reply) Thank you for your comment. As you pointed out, I have added the severe reference in the Methods section.

Results

No comment

Discussion

One topic might be stated and added to the discussion section: which common risk factor and its mechanisms might be responsible for the acquired findings?

Reply) Thank you for your valuable suggestion. In response to your comment, we have added a discussion on common risk factors (unhealthy lifestyle habits: decreased physical activity, increased carbohydrate intake) that might influence the findings observed in our study. Additionally, we have provided a separate paragraph explaining systemic inflammation as a mechanism that links periodontitis and metabolic syndrome.

“In this study, it was observed that individuals with severe periodontitis tend to exercise less regularly, have higher energy intake, and intake more carbohydrates compared to those without periodontitis (table 1). These lifestyle habits can increase the risk of metabolic syndrome, such as obesity and diabetes [32]. The increase in met-abolic syndrome, in turn, can exacerbate systemic inflammation, which may further worsen periodontitis. Several studies have also confirmed that the incidence of perio-dontitis is higher in individuals who exercise less [33,34] and intake more carbohy-drates [35]. Based on this evidence, it can be concluded that adopting a healthy lifestyle not only improves metabolic syndrome but may also help in managing periodontitis.”

                           “Another possible mechanism is that periodontitis, as a complex, chronic inflam-matory condition, can lead to a worsening of the metabolic phenotype. This inflam-matory response is marked by an abnormal secretion of host-derived factors that con-tribute to inflammation and tissue degradation. Periodontitis and adverse metabolic phenotypes share the underlying mechanism of chronic inflammatory diseases. meta-bolically abnormalities induce various proinflammatory effects, including the release of TNF-α, IL-6, and leptin, which affect multiple body systems, including the oral cav-ity [21]. D'Aiuto et al. demonstrated that chronic inflammation, a hallmark of metabol-ic disorders, can initiate processes leading to reduced macrophage and neutrophil functions, advanced glycosylation accumulation, and inflammation, all of which con-tribute to the development of periodontitis [42]. Additionally, periodontitis and ad-verse metabolic phenotypes share common risk factors such as lower income, low ed-ucation levels, less exercise, less frequent tooth brushing, and high carbohydrate intake. These shared poor lifestyle habits are likely to be risk factors contributing to the de-velopment of both periodontitis and metabolic phenotypes.”

Please, could you deliberate on the values and significant impact of the current study in order to help resolve oral health issues and systematic metabolic conditions for specific population groups?

Reply) Thank you for your insightful comment. We have carefully considered your request to elaborate on the values and significant impact of our study. In response, we have revised the second, third paragraph of the Discussion section to better articulate the implications of our findings as follows.

“This study revealed that the prevalence of periodontitis increased across the MAO, MANW, and MHO phenotype groups, compared with the MANW phenotype group, This association suggests that periodontitis may serve as a marker for metabolic syn-drome and other systemic inflammatory conditions, thereby providing a valuable op-portunity for early intervention. These findings are consistent with results from sever-al other studies, which have revealed increased prevalence rates of periodontitis in pa-tients with diabetes [15,21,22], metabolic syndrome[2], and obesity [9,16]. The recogni-tion of periodontitis as a potential indicator of broader systemic issues reinforces the importance of routine periodontal assessments as part of comprehensive health evalu-ations, particularly for those in high-risk phenotypic categories, particularly those in the MAO and MANW phenotype.”

             “Interestingly, our study found that the MANW phenotypes are more strongly as-sociated with severe periodontitis compared to the MHO phenotype. The underlying mechanisms that could account for the varying risk of periodontitis in the MANW or MHO phenotypes are not well understood. However, it is notable that some studies suggest that individuals with the MANW phenotype may have worse clinical out-comes than those with the MHO phenotype or the general population. For example, a prospective study from Korea reported that MANW phenotype showed increased ar-terial stiffness and carotid intima-media thickness compared to MANW phenotype [23]. Similarly, another prospective study from Korea reported that MANW phenotype had an unfavorable inflammatory and atherogenic profile, including higher levels of inflammatory markers, smaller LDL particles, and oxidized LDL compared to MHO phenotype [24]. In contrast, individuals with the MHO phenotype may have higher adiponectin [25] and lower levels of systemic inflammation [26] compared to MAO, leading to a weaker association with severe periodontitis. This could be due to higher levels of systemic inflammation in MANW individuals which may contribute to their increased susceptibility to severe periodontitis[27]. In contrast, individuals with the MHO phenotype, who have obesity without metabolic syndrome, may have higher adiponectin [25] and lower levels of systemic inflammation [26] compared to MAO, leading to a weaker association with severe periodontitis. Individuals with the MANW phenotype often appear non-obese, leading to delayed diagnosis. However, this group has a high prevalence of diabetes and poor long-term prognosis due to delayed diag-nosis [28,29]. Screening for metabolic syndrome in patients with periodontitis, even if they are believed to have no underlying conditions, could aid in the early detection of MANW phenotype.”

References

This section lacks sufficient references to support some of the ideas in the introduction and materials and methods sections.

Reply) Thank you for your comment. As you pointed out, I have added the severe reference in the Introduction and Methods section.

Journal names should be abbreviated. Please follow the ACS style guide; it is recommended for IJERPH.

Reply) Thank you for your insightful comment. We have revised the references using EndNote with the MDPI ACS Journals style (https://endnote.com/downloads/styles/mdpi/)

Best regards,

Reviewer 4 Report

Comments and Suggestions for Authors

The authors aim to explore the association between periodontitis and different body phenotypes, using data from the 2013-2015 Korea National Health and Nutrition Examination Survey. Compared to previous studies that focused solely on the relationship between periodontitis and either metabolism or obesity, the innovation of this study lies in using body phenotypes as the grouping criterion, combining obesity with metabolic health status. The findings demonstrate a significant association between periodontitis and adverse metabolic phenotypes, highlighting the importance of incorporating periodontal health into overall health management.

However, I see that

1.The depth and logical structure of the background introduction could be further strengthened. The previous studies on the relationship between periodontitis, metabolism, and obesity have not been detailed, and should be further supplemented.

2.The article details are not rigorous. For example, 

Where is CPI =4 ?if there are mistakes due to carelessness, please explain and clarify. And provide a more detailed explanation in the methods section.

3.In the discussion section, there is a lack of analysis regarding the comparative results of different groups (Result 3.3.). It should explain why certain body types or phenotypes have a stronger and more evident association with periodontitis, and what the possible underlying mechanisms are. This will help readers understand the scientific significance behind the data.

Comments on the Quality of English Language

Although the document contains no significant spelling errors, there are areas where the English can be further refined:

Consistency in Terminology: The document uses both “metabolically abnormal obese (MAO)” and “metabolic abnormal obese (MAO)” interchangeably. This inconsistency can be confusing to the reader. It is advisable to choose one term and use it consistently throughout the text.

Sentence Length and Readability: Some sentences are quite long, which can make them difficult to read and understand. For instance, complex sentences with multiple clauses may benefit from being split into shorter, more concise sentences. This change would improve the overall readability of the document.

Author Response

Journal Revision Letter (Response to Reviewers)

Aug 31 2024

Response to Editor and Reviewers:

Thank you for the insightful and thorough review of our manuscript and your comments and suggestions. It is with excitement that I re-submit to you our revised version of manuscript entitled “Effect of periodontitis on body size phenotypes in apparently healthy adults: the Korean National Health and Nutrition Ex-amination Survey 2013-2015” for the IJERPH. We have revised our paper, taking your comments and suggestions into account. In the process, we believe that this paper has been significantly improved.

             We have responded specifically to each suggestion below.

Sincerely,

Sang Yong Kim, MD, PhD:

Department of Endocrinology and Metabolism, Chosun University Hospital,

588 Seoseok-Dong, Dong-Gu, Gwangju, Republic of Korea.

e-mail: diabetes@chosun.ac.kr

Office: 82-62-220-3011

ORCID: 0000-0002-3902-622X

Reviewer #4

Comments and Suggestions for Authors

The authors aim to explore the association between periodontitis and different body phenotypes, using data from the 2013-2015 Korea National Health and Nutrition Examination Survey. Compared to previous studies that focused solely on the relationship between periodontitis and either metabolism or obesity, the innovation of this study lies in using body phenotypes as the grouping criterion, combining obesity with metabolic health status. The findings demonstrate a significant association between periodontitis and adverse metabolic phenotypes, highlighting the importance of incorporating periodontal health into overall health management.

However, I see that

1.The depth and logical structure of the background introduction could be further strengthened. The previous studies on the relationship between periodontitis, metabolism, and obesity have not been detailed, and should be further supplemented.

Reply) Thank you for your insightful comment. As you pointed out, we have revised the introduction to strengthen the depth and structure, especially in the relationship between periodontitis, metabolism, and obesity as follows.

             “Metabolic syndrome is defined as a cluster of conditions including hypertension, dyslipidemia, hyperglycemia, and abdominal obesity, all of which significantly in-crease the risk of cardiovascular diseases and diabetes [3]. The connection between periodontitis and metabolic syndrome is thought to be mediated by chronic systemic inflammation. Inflammatory mediators such as C-reactive protein (CRP), interleukin-6 (IL-6), and tumor necrosis factor-alpha (TNF-α) are elevated in both periodontitis and metabolic syndrome, suggesting a bidirectional relationship where each condition may exacerbate the other [2,4-6]. The presence of periodontitis may serve as a marker for systemic inflammatory burden, highlighting the importance of oral health in the man-agement and prevention of broader systemic health issues [4,7,8]. Similarly, Obesity, which is often a component of metabolic syndrome, is known to exacerbate periodon-tal disease due to increased inflammatory burden and altered immune responses [9]. Adipose tissue, particularly in obese individuals, secretes pro-inflammatory cytokines that can contribute to the systemic inflammation observed in periodontitis [10].”

2.The article details are not rigorous. For example,

Where is CPI =4 ?if there are mistakes due to carelessness, please explain and clarify. And provide a more detailed explanation in the methods section.

Reply) Thank you for pointing my manuscript error. As you point out, we have made the following revisions:

“Periodontal status was categorized as severe periodontitis (CPI = 4), mild periodontitis (CPI = 3), or no periodontitis (CPI ≤ 2 in all sextants).”

3.In the discussion section, there is a lack of analysis regarding the comparative results of different groups (Result 3.3.). It should explain why certain body types or phenotypes have a stronger and more evident association with periodontitis, and what the possible underlying mechanisms are. This will help readers understand the scientific significance behind the data.

 Reply) Thank you for your valuable comment. As you mentioned, there was a lack of supporting evidence for the comparison between phenotypes in this paper. Therefore, we have revised the 2nd and 3rd paragraphs of the Discussion section to emphasize the phenotypes, particularly the MANW phenotype, and have supplemented the discussion with additional reference studies on phenotype comparisons as follows.

“This study revealed that the prevalence of periodontitis increased across the MAO, MANW, and MHO phenotype groups, compared with the MANW phenotype group, This association suggests that periodontitis may serve as a marker for metabolic syn-drome and other systemic inflammatory conditions, thereby providing a valuable op-portunity for early intervention. These findings are consistent with results from sever-al other studies, which have revealed increased prevalence rates of periodontitis in pa-tients with diabetes [15,21,22], metabolic syndrome[2], and obesity [9,16]. The recogni-tion of periodontitis as a potential indicator of broader systemic issues reinforces the importance of routine periodontal assessments as part of comprehensive health evalu-ations, particularly for those in high-risk phenotypic categories, particularly those in the MAO and MANW phenotype.”

             “Interestingly, our study found that the MANW phenotypes are more strongly as-sociated with severe periodontitis compared to the MHO phenotype. The underlying mechanisms that could account for the varying risk of periodontitis in the MANW or MHO phenotypes are not well understood. However, it is notable that some studies suggest that individuals with the MANW phenotype may have worse clinical out-comes than those with the MHO phenotype or the general population. For example, a prospective study from Korea reported that MANW phenotype showed increased ar-terial stiffness and carotid intima-media thickness compared to MANW phenotype [23]. Similarly, another prospective study from Korea reported that MANW phenotype had an unfavorable inflammatory and atherogenic profile, including higher levels of inflammatory markers, smaller LDL particles, and oxidized LDL compared to MHO phenotype [24]. In contrast, individuals with the MHO phenotype may have higher adiponectin [25] and lower levels of systemic inflammation [26] compared to MAO, leading to a weaker association with severe periodontitis. This could be due to higher levels of systemic inflammation in MANW individuals which may contribute to their increased susceptibility to severe periodontitis[27]. In contrast, individuals with the MHO phenotype, who have obesity without metabolic syndrome, may have higher adiponectin [25] and lower levels of systemic inflammation [26] compared to MAO, leading to a weaker association with severe periodontitis. Individuals with the MANW phenotype often appear non-obese, leading to delayed diagnosis. However, this group has a high prevalence of diabetes and poor long-term prognosis due to delayed diag-nosis [28,29]. Screening for metabolic syndrome in patients with periodontitis, even if they are believed to have no underlying conditions, could aid in the early detection of MANW phenotype.”

Comments on the Quality of English Language

Although the document contains no significant spelling errors, there are areas where the English can be further refined:

Consistency in Terminology: The document uses both “metabolically abnormal obese (MAO)” and “metabolic abnormal obese (MAO)” interchangeably. This inconsistency can be confusing to the reader. It is advisable to choose one term and use it consistently throughout the text.

Reply) Thank you for pointing out this issue. We have standardized the terminology and will use 'metabolically abnormal obese (MAO)' consistently throughout the document.

Sentence Length and Readability: Some sentences are quite long, which can make them difficult to read and understand. For instance, complex sentences with multiple clauses may benefit from being split into shorter, more concise sentences. This change would improve the overall readability of the document.

Reply) Thank you for your valuable feedback. We appreciate your suggestion regarding sentence length and readability. In response, we have reviewed the document and revised the sentences that were particularly long or complex. We have split these sentences into shorter, more concise sentences to improve the overall readability of the text.

Round 2

Reviewer 1 Report

Comments and Suggestions for Authors

-